# Epidemiology and Survival of Dogs Diagnosed with Splenic Lymphoid Hyperplasia, Complex Hyperplasia, Stromal Sarcoma and Histiocytic Sarcoma

**DOI:** 10.3390/ani12080960

**Published:** 2022-04-08

**Authors:** Cleide H. Spröhnle-Barrera, Jayne McGhie, Rachel E. Allavena, Helen C. Owen, Chiara Palmieri, Tamsin S. Barnes

**Affiliations:** School of Veterinary Science, University of Queensland, Gatton, QLD 4343, Australia; j.mcghie@uq.edu.au (J.M.); r.allavena@uq.edu.au (R.E.A.); h.owen1@uq.edu.au (H.C.O.); c.palmieri@uq.edu.au (C.P.); t.barnes@uq.edu.au (T.S.B.)

**Keywords:** spleen, fibrohistiocytic nodules, lymphoid hyperplasia, complex hyperplasia, dog, sarcoma, survival

## Abstract

**Simple Summary:**

Dogs are frequently diagnosed with nodular lesions affecting the spleen. One of the most common lesions has been traditionally classified as fibrohistiocytic nodules but has more recently been separated into different types: lymphoid hyperplasia, complex hyperplasia, stromal sarcoma, and histiocytic sarcoma. Notably, there are contradictory reports on the survival times of dogs diagnosed with stromal sarcoma and histiocytic sarcoma. In addition, some studies have found an association between the diagnosis of these nodules with the dog’s age, breed, sex, and survival time. This paper aims to estimate the frequency of the previously known fibrohistiocytic nodules among splenectomized dogs and identify associations between the four different types of nodules with the age, sex, breed, and survival. Typical survival times varied markedly between the four types of nodules. These findings reveal widely different outcomes for dogs diagnosed with each fibrohistiocytic nodule, providing useful information to clinicians on the survival estimates of these lesions.

**Abstract:**

Canine splenic fibrohistiocytic nodules traditionally encompassed benign lymphoid hyperplasia, complex hyperplasia, and malignant fibrous histiocytoma. The latter has been recently re-classified into histiocytic sarcoma and stromal sarcoma. Reliable indicators of post-splenectomy survival and demographic factors predisposing to the four types of nodules are not completely understood. This study aims to estimate frequency, survival times, and identify risk factors of splenectomized dogs diagnosed with lymphoid hyperplasia, complex hyperplasia, histiocytic sarcoma, and stromal sarcoma using medical records containing histopathological diagnosis from the VetCompass Australia database (1989–2018), which collects demographic, and clinical information from veterinary clinics. Out of 693 dogs, 315 were diagnosed with fibrohistiocytic nodules, mostly lymphoid hyperplasia (169/693, 24.4%), followed by stromal sarcoma (59/693, 8.5%), complex hyperplasia (55/693, 7.9%), and histiocytic sarcoma (32/693, 4.6%). Dogs aged 8–10 years were more likely to be diagnosed with histiocytic or stromal sarcoma than lymphoid hyperplasia. Dogs diagnosed with lymphoid hyperplasia had a longer survival time than those with other diagnoses (median > 2 years). Dogs diagnosed with histiocytic sarcoma had longer survival times (median 349 days) than stromal sarcoma (median 166 days). Results suggest that knowledge of the type of splenic fibrohistiocytic nodule, patients’ age, and sex can be used to increase prognostic accuracy.

## 1. Introduction

Splenic conditions are a significant cause of morbidity and mortality in middle-aged and older dogs [1]. For this reason, extensive research has been conducted to understand the biological behavior and prognosis of dogs diagnosed with splenic disorders, especially malignant tumors such as hemangiosarcoma and histiocytic sarcoma [2,3]. In one of the first studies (1968) describing splenic nodules in necropsied dogs, lymphoid hyperplasia was the most represented lesion (27/313, 8.6%), followed by lymphosarcoma or lymphoma (11/313, 3.5%) [4]. That trend has changed over the last decades, with splenic hemangiosarcoma identified as the most common diagnosis in dogs presented with acute non-traumatic haemoabdomen in a 2003 study (21/30, 70%) [5]. Moreover, an analytical study in Brazil also noted that more than half of the splenic neoplastic conditions were diagnosed as haemangiosarcoma (9/16, 56.2%) [6]. A 2021 retrospective study found that splenic lymphoid hyperplasia was the most frequently diagnosed condition in splenectomized dogs (15/44, 34%), and haemangiosarcoma the most frequent malignant neoplasia (5/44, 11%), whereas lymphoma was less frequent (1/44, 2.3%) [7].

Splenic fibrohistiocytic nodules are nodular proliferations composed of distinct fibroblastic and histiocytic cell types that displace normal splenic architecture [8]. Spangler et al. [9] proposed a 3-tier grading system to classify the fibrohistiocytic nodules according to morphological characteristics and biological behavior. Grade 1, also known as lymphoid hyperplasia, is considered a non-neoplastic growth characterized by increased size and number of the lymphoid white pulp (>70%), with scant mesenchymal stroma and absence of necrosis [9]. Grade 2, or complex hyperplasia, represents a transitional form between benign and malignant growth, with the main morphological feature being the presence of a slight decrease of the lymphoid white pulp and an increased number of “fibrohistiocytic cells” [9]. Grade 3, known as malignant fibrous histiocytoma, features a lower proportion of lymphoid cells (<40%) with an increased proportion of “fibrohistiocytic cells” with a higher mitotic index (>10 per HPF 40×) and malignant morphologic features such as anisocytosis; in addition, necrosis is present [9]. In this study, the 3-tier classification has also suggested a progression from lymphoid hyperplasia (grade 1) and complex hyperplasia (grade 2) to malignant fibrous histiocytoma (grade 3) [9].

More recently, a reclassification of the fibrohistiocytic nodules has expanded the proposed progression between benign lymphoid hyperplasia and complex hyperplasia to lymphoma [10]. In addition, malignant fibrous histiocytoma was separated into histiocytic sarcoma and stromal sarcoma, both malignant types but associated with different survival periods [9,10]. These visceral soft tissue sarcomas are often highly heterogenous and can be difficult to differentiate without histopathology [11]. Splenic stromal sarcoma is composed of polygonal to spindle-shaped cells, with occasional multinucleated cells, large areas of necrosis, and frequent extramedullary hematopoietic cells [10]. Splenic histiocytic sarcoma consists of a pleomorphic population of round to spindle-shaped cells, with multinucleated giant cells, often accompanied by a sparse population of normal lymphocytes [10]. However, these morphological features are not always sufficient to allow for differentiation, warranting immunohistochemistry to provide a more accurate diagnosis [10,12,13]. Therefore, the standard recommendation for diagnosing splenic conditions continues to be histopathology following splenectomy or necropsy [14].

Clinically, the relative contribution of demographic factors associated with the diagnosis of the fibrohistiocytic nodules could provide further prognostic information but requires further investigation. Different studies have been conducted on splenic nodules in splenectomized dogs to determine their possible association with risk factors. A descriptive study on 224 splenectomized patients had 212 diagnosed with fibrohistiocytic nodules, with most patients (165/212, 77.8%) identified as geriatric dogs with an average age of 10.1 years [15]. Most previous publications concur that old age (10 to 11 years of age) is a common risk factor for developing the splenic condition in dogs [2,16,17,18]. In terms of breed, splenic conditions appear to be more common in several specific breeds of dogs (e.g., Labrador retrievers or German shepherd) or mixed breeds [14]. It is also well known that certain pedigree breeds are predisposed to histiocytic sarcoma, notable the Brenes mountain dog for the disseminated form and the flat-coated retriever for the localized form [15]. Likewise, few studies have considered sex as one of the predisposing factors, with male patients being affected in 40% [19] to 50% [17] of cases and females patients in 47% [18] to 57.1% [19]. In the majority of the studies [17,18,19], splenic nodular lesions have been observed with the highest frequency in castrated animals, although the castration status as a risk factor for the occurrence of tumors in domestic animals is highly dependent on the geographical distribution of the canine populations (frequency of castration, remote vs. urban areas). Furthermore, there is variability in the reported demographic factors to confidently identify the population at risk. Thus, increased knowledge of the risk factors, epidemiology, and prognosis of these nodules is likely to improve the clinical management of these patients.

Importantly, the survival of splenectomized dogs diagnosed with fibrohistiocytic nodules has only been reported in a few studies. In the 2012 study by Moore et al. [10], of 32 dogs diagnosed with one of the four types of fibrohistiocytic nodules, the median survival for lymphoid hyperplasia was 570 days, for complex hyperplasia was 387 days, and 488 days for stromal sarcoma, whereas histiocytic sarcoma had only 74 days median survival time. More recently, contradictory findings regarding survival in histiocytic sarcoma have emerged. A 2019 study that focused on clinical outcomes in 14 dogs diagnosed with splenic histiocytic sarcoma reported a median survival time of 427 days [20]. These findings suggest that despite research indicating that primary splenic neoplastic disease in dogs is associated with decreased survival time [19], accurate prognostication of the survival of dogs diagnosed with splenic fibrohistiocytic nodules is required.

Therefore, the purpose of this study is to estimate the frequency, identify risk factors, and estimate survival times of splenectomized dogs diagnosed with splenic lymphoid hyperplasia, complex hyperplasia, histiocytic sarcoma, and stromal sarcoma, using a large retrospective dataset.

## 2. Materials and Methods

### 2.1. Preliminary Screening

The Vet Compass Australia database [21], containing the electronic patient records (EPR) of dogs that had attended 96 urban and rural primary-care veterinary clinics in New South Wales, Victoria, and Queensland in Australia from 1989 to 2018, was searched using the following keywords: spleen, fibrohistiocytic, nodules, hyperplasia, histiocytic, malignant, and sarcoma. The EPRs for all patients identified in this preliminary search were compiled in an Excel spreadsheet (Microsoft Corp) file containing one row per consultation per patient, with columns for sex, breed, date of birth and death, consultation dates, and free text clinical notes for all the dogs. These represented 19,940 rows in the database, indicating a potential maximum of 19,940 veterinary consultation events. It was noted that a single visit to a veterinarian often resulted in the generation of multiple rows for a single patient. Duplicate entries and entries that could not be linked to a specific patient were deleted by selecting repeated columns and rows and then selecting ‘Remove Duplicate’ within the Data Tools tab in the spreadsheet. Multiple versions of the spreadsheet were made throughout each step, and manual cross-checks were made to maintain the accuracy of the information. After excluding duplicates, 14,865 rows remained. Data were then consolidated in a separate spreadsheet in a one-row-per-patient format.

### 2.2. Patient Identification

Patients identified in the preliminary search were assessed against inclusion and exclusion criteria. Patients were included if they had undergone splenectomy and a definitive diagnosis was made using histopathology of the spleen from biopsy or necropsy. Patients were excluded if the diagnosis was made exclusively by ultrasound and/or cytology, the diagnosis was made in other organs (e.g., cutaneous histiocytic sarcoma), and/or if the patient was euthanized during/after surgery without a final diagnosis.

### 2.3. Data Management

The data for all patients meeting the inclusion criteria were restructured for statistical analysis. The diagnosis for each patient was categorized using the histopathology data recorded in free text in the clinical notes. Keywords used to classify the four fibrohistiocytic nodule diagnoses are shown in Table 1.

The breed was categorized by grouping dogs into purebred types as defined by the Australian National Kennel Council (Toys, Terriers, Gundogs, Hounds, Working dogs, Utility, Non-Sporting) [22]. Sex and desexing status were combined to form a four-category variable: entire female, neutered female, entire male, and neutered male. Age was categorized as <8 years, 8–<10 years, and ≥10 years. Other variables were created to capture the date of splenectomy, age at splenectomy, if they were known to be deceased, cause of death, days from splenectomy to death (if deceased), days from splenectomy to the last consult (if not deceased), and treatment.

### 2.4. Statistical Analysis

In this study, the baseline population consists of splenectomized dogs diagnosed with the previously known fibrohistiocytic nodules. The relative frequencies of splenic lymphoid hyperplasia, complex hyperplasia, stromal sarcoma, and histiocytic sarcoma diagnoses among splenectomized dogs with splenic fibrohistiocytic nodules were estimated overall and by demographic categories. The risk factors and survival were analyzed by comparing the different types of fibrohistiocytic nodules. A multinomial logistic regression model was fitted to evaluate the non-linear association between gender, age, breed, and diagnosis of each type of splenic fibrohistiocytic nodule. To estimate the survival time of patients diagnosed with each type of splenic fibrohistiocytic nodule, a Kaplan-Meier survival curve for a two-year period post-splenectomy was created based on the time of death for any reason after splenectomy. If the time of death was unknown, the patient was censored at the date of last consult. A single Cox proportional hazard regression model was fitted to examine possible associations between diagnosis, sex, breed, age, and survival. The survival estimates incorporating uncertainty and adjusting for the other possible risk factors are presented in the results from the Cox model, including the adjusted hazard ratios and the confidence intervals. The assumption of proportional hazards was tested.

Stata 16^®^ (StataCorp. 2019. *Stata Statistical Software: Release 16*. StataCorp LLC: College Station, TX, USA) was used for all statistical analyses.

## 3. Results

### 3.1. Descriptive Results

A total of 15,183 patients were identified in the initial search. Of these, 795 patients were identified to have a clinical presentation of splenic condition, of which 693 had undergone splenectomy and had a histopathology-confirmed diagnosis (Figure 1). Of these, only 7 patients were diagnosed prior to 2000, all of which were hemangiosarcoma.

The results for all splenectomised dogs with a histopathology diagnosis are shown in Table 2. In most patients (685/693, 98.9%), the diagnosis was made by biopsy and only a small number of patients (8/693, 1.2%) by necropsy. There were 315 patients diagnosed with fibrohistiocytic nodules (315/693, 45.4%), and 378 diagnosed with non-fibrohistiocytic nodules (378/693, 54.4%). Of the non-fibrohistiocytic nodules, the most common diagnosis was hemangiosarcoma (258/693, 37.2%). Amongst the fibrohistiocytic nodules, most of the patients were diagnosed with benign lymphoid hyperplasia (169/693, 24.4%), followed by stromal sarcoma (59/693, 8.5%), complex hyperplasia (55/693, 7.9%), and histiocytic sarcoma (32/693, 4.6%). Additionally, among all dogs diagnosed with splenic conditions, only a small number of patients (32/693, 4.6%) were known to have received post-splenectomy treatment associated with the splenic disease.

Demographic data of patients diagnosed with fibrohistiocytic nodules are shown in Table 3. The most common age category at diagnosis was over 10 years old (189/315, 60%), followed by dogs between 8–10 years (74/315, 23.5%). The frequency of diagnosis was higher in both neutered male (113/315, 35.8%) and female dogs (111/315, 35.2%) compared to entire male (61/315, 19,4%) and entire female dogs (30/315, 9.5%). In terms of distribution of dog breeds, the groups with the highest frequency of diagnosis of splenic disease included terrier dogs and gundogs (61/315, 19.4% each), followed by toy dogs and working dogs (51/315, 25.4% each). The first case of lymphoid hyperplasia was diagnosed in 2000, histiocytic sarcoma in 2002, and both complex hyperplasia and stromal sarcoma in 2003.

### 3.2. Risk Factors for Type of Splenic Fibrohistiocytic Nodule Diagnosis

The results from the multinomial analysis are shown in Table 4. There was an association between age and diagnosis (*p* = 0.01). Specifically, dogs aged 8–<10 years were much more likely to be diagnosed with histiocytic sarcoma than lymphoid hyperplasia compared to those aged less than 8 years (Relative risk ratio [RRR]: 8.4 (95% confidence interval [CI] 1.7–41.0) and more likely to be diagnosed with stromal sarcoma (RRR: 2.8 (95% CI 0.9–8.5)) rather than lymphoid hyperplasia. Overall, there was no strong association between sex and diagnosis (*p* = 0.41), but there was some evidence that entire males were less likely than neutered males to be diagnosed with complex hyperplasia (RRR: 0.3 (95% CI 0.1–1.0)) compared to lymphoid hyperplasia. There was no evidence of a strong association between breed and diagnosis of each type of splenic fibrohistiocytic nodule (*p* = 0.91).

### 3.3. Survival Analysis

From the 315 patients with fibrohistiocytic nodular disease, 136 dogs died, and 179 were censored. Dogs diagnosed with lymphoid hyperplasia had a median survival time of at least 1095 days post-splenectomy, compared to 526 days for complex hyperplasia, 349 days for histiocytic sarcoma, and 166 days for stromal sarcoma (Figure 2).

The results from the Cox proportional hazards model are shown in Table 5. There was a strong association between the type of splenic fibrohistiocytic nodule and survival, with all other diagnoses being associated with shorter survival than a diagnosis of lymphoid hyperplasia (stromal sarcoma hazard ratio [HR]: 4.0 (95% CI 2.3–7.0), histiocytic sarcoma HR: 2.5 (95% CI 1.3–5.1), complex hyperplasia HR: 2.0 (95% CI 1.1–3.8)). Survival was also shorter in dogs aged ≥10 years compared to 8–<10 years (HR: 2.0 (95% CI 1.1–3.6)), and there was some evidence that survival was shorter in neutered females compared to neutered males (HR: 1.9 (95% CI 1.1–3.1)). No evidence of a violation of the proportional hazard assumption was found.

## 4. Discussion

The results of this large retrospective study suggest that lymphoid hyperplasia is the second most frequently diagnosed splenic condition in splenectomised dogs and that the outcome of dogs diagnosed with histiocytic sarcoma might be prognostically better than previously reported. Furthermore, there was strong evidence of an association between the diagnosis type and survival, as well as the age of the patients. Our study overcomes the limitation of the small sample size of previous publications evaluating the association between tumor diagnosis and survival, providing additional epidemiologic and prognostic insights on these splenic lesions in dogs.

Fibrohistiocytic nodules were commonly diagnosed among splenectomized dogs (315/693, 45.5%). Most of these dogs were diagnosed with benign lymphoid hyperplasia (169/315, 53.6%). These results corroborate the findings of previous work, including a 2019 analytical study, which reported lymphoid hyperplasia as the most common benign nodule diagnosed among splenectomized dogs (27/125; 21.6%) [17]. In our analysis, among dogs diagnosed with fibrohistiocytic nodules, the most frequent malignant nodules were stromal sarcoma (59/315, 18.7%), and less frequently histiocytic sarcoma (32/315, 10.1%). These results are similar to those found in a study of 32 splenectomized patients diagnosed with fibrohistiocytic nodules, where the most common malignant nodule was stromal sarcoma (8/32, 25%), followed by histiocytic sarcoma (6/32, 18.8%) [10]. In contrast, these findings are in disagreement with an analytical study that estimated the frequency of benign versus malignant conditions in splenectomized dogs, suggesting that histiocytic sarcoma (16/325, 4.9%) is more frequently diagnosed than stromal sarcoma (3/325, 0.9%) [23]. These differences can be partially explained by different methodologies used in each research study. For instance, most of the previous studies gathered their data from referring hospitals, while this study focused on primary care general practice veterinary clinics. In addition, most of the aforementioned studies focused on comparing the frequency of benign versus malignant splenic conditions overall [15,17,23,24,25], while this study focused on the frequency of fibrohistiocytic nodules only [8,9,10,26].

Several different risk factors have been considered in previous publications on canine splenic conditions, including age, breed, sex, and total body weight [12,17,25]. Most studies included animals of all ages, reporting older dogs (median 8–12 years) were more likely to be diagnosed with splenic tumors compared to younger dogs [15,17,18,23,24,27,28,29,30]. This study found a higher frequency of diagnosis in dogs over 10 years old (189/315, 60%). It also clarified that dogs aged 8–<10 years were much more likely than those aged less than 8 years to be diagnosed with histiocytic sarcoma compared to lymphoid hyperplasia and more likely to be diagnosed with stromal sarcoma rather than lymphoid hyperplasia. Thus, age range is important when considering splenectomy patients. The body weight was not considered in this study since this variable was not consistently reported in the master dataset, including the information about the scale system adopted.

In this study, neutered male (113/315, 35.8%) and neutered female dogs (111/315, 35.2%) were more frequently diagnosed with the previously known fibrohistiocytic nodules compared to entire dogs. These results differ from the study published by Spangler and Kass in 1998 [9], which proposes that female dogs (58/96, 60.4%) are more likely to be diagnosed with fibrohistiocytic nodules than male dogs (38/96, 39.5%). However, our analysis showed no evidence of a strong association overall between sex and diagnosis (*p* = 0.41), but there was some evidence that entire males were less likely than neutered males to be diagnosed with complex hyperplasia (RRR: 0.3 (95% CI 0.1–1.0)) rather than lymphoid hyperplasia. These findings raise the possibility that the sex of the patient might have an association with the type of fibrohistiocytic nodule diagnosed in splenectomized dogs.

The association between dog breed and diagnosis has mostly been described concerning the frequency of splenic conditions among mixed and pure-breed dogs [15,18,19,23]. This study has instead evaluated the frequency of splenic fibrohistiocytic nodules in dog breed groups, such as gundogs or working dogs. Surprisingly, there was no evidence of a strong association between the breed groups selected and the diagnosis of each type of splenic fibrohistiocytic nodule (*p* = 0.91). These results corroborate the findings of a retrospective study of benign and malignant splenic conditions conducted in Australia, which found a higher frequency of large breed dogs diagnosed with malignant neoplasia, but no significant prevalence for any dog breed type [24]. This indicates that most dogs are likely not genetically predisposed to histiocytic sarcoma, unlike one other study that has found a strong breed predisposition for histiocytic sarcoma diagnosis in Bernese Mountain dogs [31].

The results of the 2-year post-splenectomy survival time analysis for benign lymphoid hyperplasia (>2 years median) and complex hyperplasia (median 526 days) are similar to those previously reported. Diagnosis of complex hyperplasia, histiocytic sarcoma, and stromal sarcoma is associated with shorter survival than a diagnosis of lymphoid hyperplasia. In the 3-tier classification, diagnosis of either benign lymphoid hyperplasia or complex hyperplasia (grade 1 and 2, respectively) was associated with a probability of 87% survival 12-months post-splenectomy, while malignant fibrous histiocytoma (grade 3) was associated with 55% survival 12 months post-splenectomy [9]. The survival time as reported in our study is longer for dogs diagnosed with histiocytic sarcoma (median 349 days) but shorter for stromal sarcoma-affected patients (median 166 days) compared to Moore et al. [10], who proposed shorter survival times for dogs diagnosed with histiocytic sarcoma (median 74 days), and longer for stromal sarcoma (488 days). These discrepancies are likely associated with population size, with only 27 of 32 patients with fibrohistiocytic splenic nodules reviewed in the 2012 publication [10] compared to the 315 patients in this study. Moreover, a study by Latifi et al. provided further support to our results, with a long survival time for splenectomized dogs diagnosed with localized splenic histiocytic sarcoma (median 427 days, *n* = 14) [20]. Similar results are reported in another study, with localized histiocytic sarcoma showing an overall survival time of more than one year (median 398 days, *n* =180) [32].

There were several limitations in this study. First, a risk of misclassification of the lesions due to the different histological terminology applied to the same type of nodules by different pathologists. Also, unlike other publications, this study was based on cancer records without the opportunity to review histological samples or further confirm the diagnosis by immunohistochemistry. Furthermore, because some dogs were excluded if diagnosed by ultrasound and cytology alone, our results might underestimate the true relative frequency of benign splenic nodules commonly found during routine health checks in asymptomatic dogs. Another limitation was the lack of post-splenectomy treatment information, making it impossible to assess improvement in survival due to treatment type or duration. Lastly, clinical records were not exhaustive regarding specific causes of death. Thus, it is not known whether the animals included in this study had died from the splenic nodules.

However, this retrospective study reports on the largest number of patients diagnosed over a long study period. In addition, restricting the study to splenectomized dogs with a definitive diagnosis allowed reliable identification of patients diagnosed with fibrohistiocytic nodules. Furthermore, the study population included patients visiting veterinary clinics rather than referral hospitals, which provides relevant data for primary care veterinarians.

## 5. Conclusions

The findings reported in our study have significant implications for understanding the behavior of the different types of fibrohistiocytic nodules, particularly survival times associated with each histological classification, and the importance of the demographic risk factors. In summary, patient age was strongly associated with the diagnosed type of fibrohistiocytic nodules, and there was a weak association with gender. This new knowledge should help clinicians improve the management of affected dogs, providing better and more accurate advice to owners regarding the prognosis of splenectomized dogs diagnosed with specific types of fibrohistiocytic nodules.

## Figures and Tables

**Figure 1 animals-12-00960-f001:**
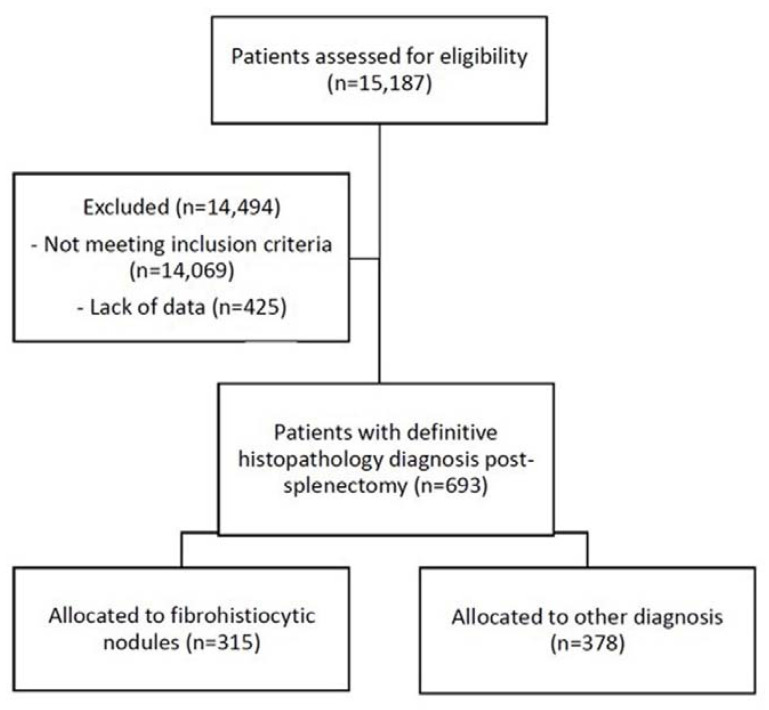
Flow diagram of the process of inclusion and exclusion of patients from the VetCompass Australia database in the study.

**Figure 2 animals-12-00960-f002:**
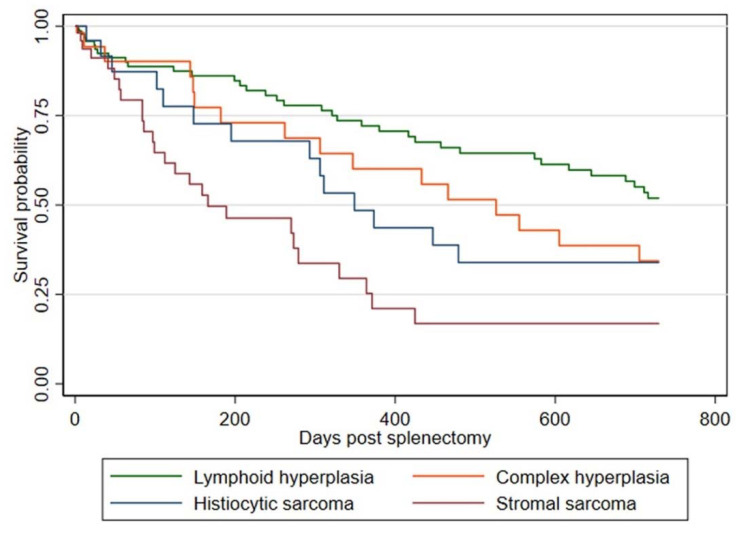
Kaplan-Meier Curve for 2-year post-splenectomy survival for each type of fibrohistiocytic nodule among dogs identified from the VetCompass Australia database that attended primary-care veterinary clinics during 1989–2018 in Australia.

**Table 1 animals-12-00960-t001:** Key histopathology terms from free-text clinical notes used to classify fibrohistiocytic nodules into the four distinct diagnoses.

Histopathological Diagnosis	Histopathological Diagnosis Extended Terminology
Lymphoid hyperplasia	Benign hyperplastic nodules/Fibrohistiocytic grade 1/Nodular hyperplasia/Lymphoid follicular hyperplasia
Complex hyperplasia	Complex nodular hyperplasia/Fibrohistiocytic grade 2/Nodular hyperplasia complex
Histiocytic sarcoma	Histiocytic/ Histiocytic sarcoma/Fibrohistiocytic grade 3/Malignant histiocytosis (splenic)
Stromal sarcoma	Stromal/Splenic Sarcoma/Sarcoma grade 3

**Table 2 animals-12-00960-t002:** VetCompass Australia database overall histopathology diagnosis in splenectomized dogs from primary-care veterinary clinics.

Variable	Category	Freq.	Percent (%)
Testing	Biopsy	685	98.9
	Necropsy	8	1.2
Diagnosis	Lymphoid hyperplasia	169	24.4
	Complex hyperplasia	55	7.9
	Histiocytic sarcoma	32	4.6
	Stromal sarcoma	59	8.5
	Hemangiosarcoma	258	37.2
	Haemangioma	4	0.6
	Haematoma	42	6.1
	Other sarcoma	3	0.4
	Lymphoma	15	2.2
	Carcinoma	3	0.4
	Lipoma	3	0.4
	Leiomyoma	1	0.1
	Myeloproliferative	13	1.9
	Normal	9	1.3
	Torsion	3	0.4
	Splenomegaly	24	3.5
Treatment	Yes	32	4.6
	No	0	0.0
	Unknown	661	95.4
Total		693	100

**Table 3 animals-12-00960-t003:** Distribution of patients diagnosed with splenic lymphoid hyperplasia, complex hyperplasia, histiocytic sarcoma, and stromal sarcoma by age, sex, and breed among splenectomized dogs with a definitive histopathological diagnosis of the four splenic nodules identified from the VetCompass Australia database covering 96 primary-care veterinary clinics between 1989 and 2018.

	Lymphoid Hyperplasia (%)	Complex Hyperplasia (%)	Histiocytic Sarcoma (%)	Stromal Sarcoma (%)	Total (%)
**Age category (years)**					
<8	33 (63.5)	11 (21.2)	2 (3.8)	6 (11.5)	52 (100)
8–<10	30 (40.5)	12 (16.2)	16 (21.6)	16 (21.6)	74 (100)
≥10	106 (56.1)	32 (16.9)	14 (7.4)	37 (19.6)	189 (100)
**Sex**					
Female entire	17 (56.7)	3 (10)	5 (16.7)	5 (16.7)	30 (100)
Female neutered	59 (53.2)	19 (17.1)	11 (9.9)	22 (19.8)	111 (100)
Male entire	40 (65.6)	7 (11.5)	6 (9.8)	8 (13.1)	61 (100)
Male neutered	53 (46.9)	26 (23)	10 (8.8)	24 (21.2)	113 (100)
**Breed ***					
Toys	24 (47.1)	8 (15.7)	8 (15.7)	11 (21.6)	51 (100)
Terriers	35 (57.4)	12 (19.7)	6 (9.8)	8 (13.1)	61 (100)
Gundogs	34 (55.7)	9 (14.8)	7 (11.5)	11 (18)	61 (100)
Hounds	19 (61.3)	4 (13)	1 (3.2)	7 (22.6)	31 (100)
Working	29 (56.8)	8 (15.7)	3 (5.9)	11 (21.6)	51 (100)
Utility	14 (41.2)	10 (29.4)	4 (11.7)	6 (17.7)	34 (100)
Non-sporting	10 (50)	3 (15)	3 (15)	4 (20)	20 (100)
**Total**	**169 (53.6)**	**55 (17.5)**	**32 (10.2)**	**59 (18.7)**	**315 (100)**

* Unknown-breed dogs were removed from the Breed analysis.

**Table 4 animals-12-00960-t004:** Multinomial analysis of risk factors for diagnosis of the different types of splenic fibrohistiocytic nodules in dogs identified from the VetCompass Australia database that attended primary-care veterinary clinics during 1989–2018 in Australia.

Diagnosis	Variable/Category	RRR (95% CI)	*p*-Value
Lymphoid hyperplasia		Reference	
Complex hyperplasia	**Age (years)**		
	<8	Reference	
	8–<10	0.9 (0.3–2.5)	0.84
	≥10	2.5 (1.3–5.1)	0.49
	**Sex**		
	Female entire	0.3 (0.1–1.3)	0.12
	Female neutered	0.6 (0.3–1.3)	0.19
	Male entire	0.3 (0.1–1)	0.04
	Male neutered	Reference	
	**Breed**		
	Toys	Reference	
	Terriers	1.3 (0.4–3.7)	0.66
	Gundogs	0.8 (0.3–2.5)	0.72
	Hounds	0.7 (0.2–2.8)	0.61
	Working dogs	0.8 (0.2–2.4)	0.62
	Utility	2.1 (0.7–6.8)	0.19
	Non-sporting	1.0 (0.2–4.7)	0.99
Histiocytic sarcoma	**Age (Years)**		
	<8	Reference	
	8–<10	8.4 (1.7–41)	0.01
	≥10	1.9 (0.4–9.6)	0.39
	**Sex**		
	Female entire	1.7 (0.4–6.0)	0.44
	Female neutered	1.2 (0.4–3.3)	0.70
	Male entire	0.8 (0.3–2.7)	0.77
	Male neutered	Reference	
	**Breed**		
	Toys	Reference	
	Terriers	0.6 (0.2–2.2)	0.47
	Gundogs	0.6 (0.2–1.9)	0.38
	Hounds	0.2 (0.01–1.5)	0.10
	Working dogs	0.4 (0.08–1.6)	0.16
	Utility	0.7 (0.2–2.9)	0.61
	Non-sporting	0.9 (0.2–4.5)	0.91
Stromal sarcoma	**Age (Years)**		
	<8	Reference	
	8–<10	2.8 (0.9–8.5)	0.05
	≥10	1.8 (0.7–4.8)	0.22
	**Sex**		
	Female entire	0.5 (0.2–1.8)	0.31
	Female neutered	0.8 (0.4–1.7)	0.70
	Male entire	0.5 (0.2–1.2)	0.14
	Male neutered	Reference	
	**Breed**		
	Toys	Reference	
	Terriers	0.5 (0.2–1.6)	0.28
	Gundogs	0.7 (0.3–1.9)	0.50
	Hounds	0.8 (0.3–2.7)	0.79
	Working dogs	0.8 (0.3–2.4)	0.80
	Utility	0.8 (0.2–2.8)	0.80
	Non-sporting	0.9 (0.2–3.7)	0.90

**Table 5 animals-12-00960-t005:** Results from Cox proportional hazard model estimating the associations between type of splenic fibrohistiocytic nodule, age, sex, and breed on survival of dogs identified from the VetCompass Australia database that attended primary-care veterinary clinics during 1989–2018 in Australia.

Variable/Category	Hazard Ratio (95% CI)	*p*-Value
**Diagnosis**		**˂0.001**
Lymphoid hyperplasia	Reference	
Complex hyperplasia	2.0 (1.1–3.8)	0.03
Histiocytic sarcoma	2.5 (1.3–5.1)	0.01
Stromal sarcoma	4.0 (2.3–7)	0.00
**Age (years)**		**0.02**
<8	0.9 (0.3–2.7)	0.90
8–<10	Reference	
≥10	2.0 (1.1–3.6)	0.02
**Sex**		**0.08**
Female entire	1.0 (0.3–3.0)	0.97
Female neutered	1.9 (1.1–3.1)	0.01
Male entire	1.3 (0.6–2.9)	0.45
Male neutered	Reference	
**Breed**		**0.47**
Toys	Reference	
Terriers	1.2 (0.6–2.4)	0.54
Gundogs	1.6 (0.8–3.2)	0.14
Hounds	1.3 (0.6–3)	0.56
Working dogs	0.9 (0.4–1.8)	0.71
Utility	1.9 (0.9–4.2)	0.10
Non-sporting	1.0 (0.3–3.2)	0.93

## Data Availability

The data presented in this study are available on request from the corresponding author.

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
