# Peer review of "Epidemiology and Survival of Dogs Diagnosed with Splenic Lymphoid Hyperplasia, Complex Hyperplasia, Stromal Sarcoma and Histiocytic Sarcoma"

_animals, 2022, doi:10.3390/ani12080960_

Round 1

Reviewer 1 Report

Review of “Epidemiology and survival of dogs diagnosed with splenic lymphoid hyperplasia, complex hyperplasia, stromal sarcoma, and histiocytic sarcoma”

The authors present a clinical sample of the various types of fibrohistiocytic nodules found in resected dog spleens from Australia, their epizootiology, and their effects on patient survival.

This is an interesting paper using a generally sound methodology. However, it could be improved by including a few additional statistical analyses, which I will outline in this review.

Not sure what the numbers in the keyword section are supposed to indicate, please remove. I would also suggest using slightly more specific keywords, as the ones chosen appear to be a bit generic.

Introduction

I’m not entirely sure why one would arbitrarily choose “traumatic injuries” as a baseline and then essentially say very little of substance – please just provide the clinical incidence, and use that to compare it to other common diseases if you want to.

Line 52, please provide the percentages as you did in line 50. Given how common lymphoma is in dogs, it would be useful to also add the numbers for this disease here.

Line 90, I would refer to old age as a “risk factor” as opposed to a “contributing factor”.

Line 91, would be worth discussing the breed specificity of histiocytic sarcoma.

Line 92, are there any data regarding desexing status as a risk factor?

M&M

 Line 127, please provide more specific information on how you identified duplicates.

Statistical analysis methods are sound in principle. However, more analyses are warranted as specified below.

Results

 Line 179 and following, I am not entirely clear on how many of these dogs were diagnosed pre-op and how many were diagnosed based on the splenectomized tissue, please clarify.

Table 3/4 would benefit from knowing what the baseline frequency for intact and desexed dogs of both sexes is in the studied population, and whether there is a significant difference between the overall population and dogs with the splenic lesions in question. Same comment for breed groups and ages.

Line 231 and following, please don’t mix years and days like that, it makes comparisons more difficult.

Figure 2, what would this look like if you added censoring indicators and/or confidence intervals?

Table 5, for the Cox model, I think it would make more sense to treat age as a continuous rather than a categorical variable in this context.

As above, there needs to be a comparison between your affected dogs and your baseline population regarding all demographic factors you considered, i.e. sex/desexing, breed group, and age distribution.

Discussion

Line 272, what was the reason you didn’t consider body weight for your models? Given its importance as a predictor of overall survival in dogs and the fact that it has been reported as a risk factor for splenic conditions, I would recommend including that in your models assuming the data are available.

Line 285 etc. – again, this is not a terribly meaningful result if you don’t compare your patient demographics to your overall population, please do so.

Line 303 and following, what if any information do we have regarding cause of death in the dogs known to have died? What percentage of mortality was attributed to the history of splenic pathology?

Author Response

Dear Editor-in-Chief and Reviewers,

Thank you for giving us the opportunity to submit a revised draft of the manuscript entitled “Epidemiology and survival of dogs diagnosed with splenic lymphoid hyperplasia, complex hyperplasia, stromal sarcoma, and histiocytic sarcoma” to Animals.

We appreciate the time and effort that you and the reviewers have dedicated to providing your valuable feedback on our manuscript. We are grateful to the reviewers for their insightful comments. We have been able to incorporate changes to reflect all the suggestions provided by the reviewers.

In addition to the reviewers’ comments, we have further revised the manuscript and the supplementary material for spelling, grammar and formatting.

We look forward to hearing from you in due time regarding our submission and to responding to any further questions and comments you may have.

Hereafter is a point-by-point response to the reviewers’ comments and concerns in bold and italic.

Sincerely,

Cleide H Sprohnle-Barrera

(on behalf of all the authors)

Reviewer #1

The authors present a clinical sample of the various types of fibrohistiocytic nodules found in resected dog spleens from Australia, their epizootiology, and their effects on patient survival.

This is an interesting paper using a generally sound methodology. However, it could be improved by including a few additional statistical analyses, which I will outline in this review.

Not sure what the numbers in the keyword section are supposed to indicate, please remove. I would also suggest using slightly more specific keywords, as the ones chosen appear to be a bit generic.

Numbers in the keyword section and keywords have been changed accordingly.

Introduction

I’m not entirely sure why one would arbitrarily choose “traumatic injuries” as a baseline and then essentially say very little of substance – please just provide the clinical incidence, and use that to compare it to other common diseases if you want to.

Thanks for pointing this out. Comparing splenic disorders with traumatic injuries as a baseline is not actually correct, since the causes of death may be variable according to the breed and age. We have reformulated the sentence and changed the reference.

Line 52, please provide the percentages as you did in line 50. Given how common lymphoma is in dogs, it would be useful to also add the numbers for this disease here.

This section has been re-phrased and percentage data has been included.

Line 90, I would refer to old age as a “risk factor” as opposed to a “contributing factor”.

Changed accordingly

Line 91, would be worth discussing the breed specificity of histiocytic sarcoma.

Included in the text now with a new reference.

Line 92, are there any data regarding desexing status as a risk factor?

Including the castrations status in epidemiological studies of risk factors in small animal cancers is very tricky, since data can be skewed by the different frequency of castration in some canine populations and countries. And this is actually quite obvious when examining the published literature on incidence of splenic lesion in dogs. It is not surprising that in all papers we have a higher incidence of those lesions in castrated females and males compared to intact animals. Fernandez: 48% spayed females, 50% neutered males, 2% intact males. Leyva et al. (out of 125 dogs): 1 intact female, 59 spayed females, 11 intact males, 54 neutered males. Cleveland and Casale (out of 105 dogs): 60 spayed females, 42 castrated males, 3 intact females.

M&M

 Line 127, please provide more specific information on how you identified duplicates.

Duplicates were located by selecting all the columns and rows containing data in the original database Excel Spreadsheet. Then selecting the Data tab, followed by Remove Duplicates from the Data Tools sections of the Spreadsheet. To maintain records and check the data kept was accurate multiple versions of the document were made at each stage, and manual cross checks were conducted between the new and old version of the dataset.

Statistical analysis methods are sound in principle. However, more analyses are warranted as specified below.

Results

 Line 179 and following, I am not entirely clear on how many of these dogs were diagnosed pre-op and how many were diagnosed based on the splenectomized tissue, please clarify.

As specified in the title and throughout the methodology, all the cases included in the study were diagnosed post-splenectomy.

Table 3/4 would benefit from knowing what the baseline frequency for intact and desexed dogs of both sexes is in the studied population, and whether there is a significant difference between the overall population and dogs with the splenic lesions in question. Same comment for breed groups and ages.

Thanks for raising this point that gives us the opportunity to clarify our methodology. The usefulness of including/excluding the castration status has been explained in the previous response. Our baseline population is not the overall population of dogs with splenic lesions. The research questions of our study are specifically about the population with one of the four nodules. We are not looking at risk factors/survival for the four nodule population relative to a very heterogeneous population of haemangiosarcomas, hematomas, torsions et etc. Rather risk factors/survival for one nodule type relative to another.

Line 231 and following, please don’t mix years and days like that, it makes comparisons more difficult.

This has been accordingly corrected in the manuscript now to display time in days.

Figure 2, what would this look like if you added censoring indicators and/or confidence intervals?

We tried different options in order to present the 4 survival curves in a way that was easily understandable by all the readers. The graph shows the crude survival pattern whereas survival estimates incorporating uncertainty and adjusting for the other possible risk factors are presented in the results from the Cox PH model (adjusted hazard ratios plus confidence intervals).

Table 5, for the Cox model, I think it would make more sense to treat age as a continuous rather than a categorical variable in this context.

Thanks for opening this discussion, although we believe that age should not be considered a continuous variable as the categorised results – particularly in the multi-nominal model – indicate a non-linear relationship. Although this could potentially have been explored in more details using polynomials the results from such models are not readily interpretable by clinicians.

As above, there needs to be a comparison between your affected dogs and your baseline population regarding all demographic factors you considered, i.e. sex/desexing, breed group, and age distribution.

Our baseline population is actually the entire population of dogs with the four nodules, as clarified above.

Discussion

Line 272, what was the reason you didn’t consider body weight for your models? Given its importance as a predictor of overall survival in dogs and the fact that it has been reported as a risk factor for splenic conditions, I would recommend including that in your models assuming the data are available.

The distribution of the body condition score and body weight data within the VetCompass Australia database is patchy at best. Moreover, some vets use a nine-point scale and other use a five-point scale but do not necessarily report which one they are reporting against. This is the main reason why we decided to take out body weight from our statistical analysis.

Line 285 etc. – again, this is not a terribly meaningful result if you don’t compare your patient demographics to your overall population, please do so.

The study is focused on comparing the 4 groups, without including the entire population of dogs affected by splenic lesions. Please see further comments above.

Line 303 and following, what if any information do we have regarding cause of death in the dogs known to have died? What percentage of mortality was attributed to the history of splenic pathology?

This is one of the limitations of our study. Unfortunately, the clinical records may not be exhaustive in terms of specific causes of death and those data were missing in most cases.

Reviewer 2 Report

The manuscript aims to estimate the epidemiology, survival times, and risk factors related to spleen lesions in dogs.

Changes in design study are necessary.

I suggest deleting total clinical records, clinical cases collected before the 2000 year and lesions diagnosed with the necroscopy and reformulating the results in consideration that I recommended above.

Also, in the discussion section, it would be interesting to evaluate and correlate the results, considering the body weight and histopathological findings.

1.What is the main question addressed by the research?

The study evaluate the frequency and the survival time of nodular lesions affecting the spleen in dogs.
2. Do you consider the topic original or relevant in the field, and if
so, why?

The topic is original, analyzing the frequency of presence of splenic nodular lesions in dogs. The scientific soundness is average.
3. What does it add to the subject area compared with other published
material?

The survival time is reported in few studies and only for any type of nodular lesions.
4. What specific improvements could the authors consider regarding the
methodology?

The authors consider the lesion diagnose by the biopsy and cytology. I suggest to evaluate only one cases diagnosed with biopsy (more correct).
5. Are the conclusions consistent with the evidence and arguments
presented and do they address the main question posed?

Yes, in my opinion
6. Are the references appropriate?

Yes, in my opinion

Author Response

Dear Editor-in-Chief and Reviewers,

Thank you for giving us the opportunity to submit a revised draft of the manuscript entitled “Epidemiology and survival of dogs diagnosed with splenic lymphoid hyperplasia, complex hyperplasia, stromal sarcoma, and histiocytic sarcoma” to Animals.

We appreciate the time and effort that you and the reviewers have dedicated to providing your valuable feedback on our manuscript. We are grateful to the reviewers for their insightful comments. We have been able to incorporate changes to reflect all the suggestions provided by the reviewers.

In addition to the reviewers’ comments, we have further revised the manuscript and the supplementary material for spelling, grammar and formatting.

We look forward to hearing from you in due time regarding our submission and to responding to any further questions and comments you may have.

Hereafter is a point-by-point response to the reviewers’ comments and concerns in bold and italic.

Sincerely,

Cleide H Sprohnle-Barrera

(on behalf of all the authors)

Reviewer #2

The manuscript aims to estimate the epidemiology, survival times, and risk factors related to spleen lesions in dogs.

Changes in design study are necessary.

I suggest deleting total clinical records, clinical cases collected before the 2000 year and lesions diagnosed with the necroscopy and reformulating the results in consideration that I recommended above.

Although patient records from 1989 were available in this retrospective study, all the splenectomized dogs diagnosed with the four types of splenic nodules (see title above) were diagnosed after the year 2000. Patients with other splenic conditions were among those diagnosed prior to the year 2000, which were not included in the analytic study of this manuscript.

Also, in the discussion section, it would be interesting to evaluate and correlate the results, considering the body weight and histopathological findings.

As mentioned in our reply to Reviewer #1, the distribution of the body condition score and body weight data within the VetCompass Australia database is patchy at best. Moreover, some vets use a nine-point scale and other use a five-point scale but do not necessarily report which one they are reporting against. This is the main reason why we decided to take out body weight from our statistical analysis, thus is not available.

1.What is the main question addressed by the research?

The study evaluate the frequency and the survival time of nodular lesions affecting the spleen in dogs.

  1. Do you consider the topic original or relevant in the field, and if
    so, why?

The topic is original, analyzing the frequency of presence of splenic nodular lesions in dogs. The scientific soundness is average.

  1. What does it add to the subject area compared with other published
    material?

The survival time is reported in few studies and only for any type of nodular lesions.

Thanks for understanding the great value of this paper in terms of providing important data on the survival rate of all the nodular lesions under investigation

  1. What specific improvements could the authors consider regarding the
    methodology?

The authors consider the lesion diagnose by the biopsy and cytology. I suggest to evaluate only one cases diagnosed with biopsy (more correct).

As mentioned in the paper: “Patients were included if they had undergone splenectomy and a definitive diagnosis was made using histopathology of the spleen from biopsy or necropsy. Patients were excluded if the diagnosis was made exclusively by ultrasound and/or cytology, the diagnosis was made in other organs (e.g., cutaneous histiocytic sarcoma), and/or if the patient was euthanized during/after surgery without a final diagnosis”

  1. Are the conclusions consistent with the evidence and arguments
    presented and do they address the main question posed?

We believe that our study is exhaustively addressing the original research questions and provide a wealth of comprehensive knowledge on a topic that ahs not been adequately addressed so far by the existing literature (especially considering the impressive amount of cases that we were able to generate).

  1. Are the references appropriate?

All the references have been rechecked, they are appropriate with the topic and new references have been included in response to reviewer #1.

Round 2

Reviewer 1 Report

Thank you for your revisions and comments. My general impression is that with the exception of the revised introduction, the authors have put much more energy into discussing my comments than they have into revising their manuscript. Please include your extensive responses to my suggestions in the revised manuscript.

Author Response

Dear Editor-in-Chief and Reviewers,

Thank you for giving us the opportunity to submit a revised draft of the manuscript entitled “Epidemiology and survival of dogs diagnosed with splenic lymphoid hyperplasia, complex hyperplasia, stromal sarcoma, and histiocytic sarcoma” to Animals.

We have included most of the comments from the previous rebuttal letter into the manuscript as suggested by reviewer #1.

We look forward to hearing from you in due time regarding our submission and to responding to any further questions and comments you may have.

Sincerely,

Cleide H Sprohnle-Barrera

(on behalf of all the authors)

Reviewer 2 Report

Well done

Author Response

Dear Editor-in-Chief and Reviewers,

Thank you for giving us the opportunity to submit a revised draft of the manuscript entitled “Epidemiology and survival of dogs diagnosed with splenic lymphoid hyperplasia, complex hyperplasia, stromal sarcoma, and histiocytic sarcoma” to Animals.

We have included most of the comments from the previous rebuttal letter into the manuscript as suggested by reviewer #1.

We would also would like to thank reviewer #2 for the time and effort given to our manuscript. 

We look forward to hearing from you in due time regarding our submission and to responding to any further questions and comments you may have.

Sincerely,

Cleide H Sprohnle-Barrera

(on behalf of all the authors)

Round 3

Reviewer 1 Report

Thank you, the paper is acceptable for publication in this form.